# The Interpersonal–Psychological Theory of Suicide in Medical Students: Comparisons of Individuals without Suicidality, Ideators, and Planners

**DOI:** 10.3390/ijerph182111526

**Published:** 2021-11-02

**Authors:** Alice Solibieda, Marianne Rotsaert, Gwenolé Loas

**Affiliations:** 1Department of Psychiatry & Laboratory of Psychiatric Research (ULB 266), Cliniques Universitaires de Bruxelles, Université Libre de Bruxelles (ULB), 1070 Bruxelles, Belgium; alice.solibieda@erasme.ulb.ac.be; 2Department of Psychology & Laboratory of Psychiatric Research (ULB 266), Department of Psychiatry, Cliniques Universitaires de Bruxelles, Université Libre de Bruxelles (ULB), 1070 Bruxelles, Belgium; marianne.rotsaert@erasme.ulb.ac.be

**Keywords:** interpersonal theory on suicide, perceived burdensomeness, thwarted belongingness

## Abstract

The aim of this study was to explore the interpersonal–psychological theory of suicide (IPTS) in medical students. Higher levels of thwarted belongingness and perceived burdensomeness were expected in medical students with suicidality compared with medical students without suicidality, and a high level of acquired capability was expected in planners compared with ideators. Recruited for the study were 178 undergraduate medical students at the Université Libre de Bruxelles (ULB): 95 subjects without suicidality, 24 subjects with lifetime suicidality, 28 subjects with recent suicidal ideation, and 26 planners. An ad hoc questionnaire evaluated the risk of suicide as well as the “Suicidal thoughts and wishes” item of the BDI-II. The Interpersonal Needs Questionnaire (INS) measured thwarted belongingness (TB) and perceived burdensomeness (PB). The Acquired Capability for Suicide Scale (ACSS) measured notably fearlessness of death or pain tolerance and depression was rated using the revised version of the Beck Depression Inventory (BDI). Cognitive–affective symptoms of depression (CA-BDI) were assessed using six items of the BDI. Analyses of variance showed significant differences between groups for TB and PB but not for ACSS. Analyses of covariance, controlling for the CA-BDI scores, confirmed the significance of differences in TB and PB. Post hoc tests showed that (1) high levels of TB were characteristic of subjects with recent suicidal ideation and planners compared with subjects without suicidality; and (2) high levels of PB were characteristic of planners compared with the three other groups. Among the three characteristics of the IPTS, PB could be a strong predictor of severe suicide risk in medical students.

## 1. Introduction

In a recent meta-analysis, the prevalence of depression or depressive symptoms among medical students was 27.2%, whereas the frequency of suicidal ideation was 11.1% [1]. The suicide rate was 5.9–8.7/100,000 medical students/year in a recent Canadian national survey-based study [2]. A recent systematic review and meta-analysis on suicide risk among healthcare workers was performed [3] and reported a standardized mortality rate of 1.44 in physicians, with a higher risk in women.

Among the different theories on suicide, the interpersonal–psychological theory of suicide (IPTS) proposed by Joiner [4] suggests that one combination of factors must be observed among ideators, planners, and attempters. Perceived burdensomeness is an individual’s miscalculation that one’s death is more beneficial to others than their continued life. Thwarted belongingness describes a sense of isolation from others and acquired capability for suicide refers to the physical capacity to inflict lethal self-harm and the fearlessness of death.

The IPTS suggests that interactions among three psychological characteristics (thwarted belongingness, perceived burdensomeness, and acquired capability) lead to suicidal ideation, plans, and attempts [4,5]. Together, continued experiences of thwarted belongingness and perceived burdensomeness, as well as hopelessness, may lead to suicidal ideation. When acquired capability is present simultaneously with thwarted belongingness and perceived burdensomeness, then an individual will make a serious suicide attempt.

The authors [5] reported that thwarted belongingness and perceived burdensomeness could differentiate individuals with suicidality from individuals without suicidality, and that acquired ability could differentiate planners and attempters from ideators. The IPTS has been tested in various groups of subjects, including university students, but, to the best of our knowledge, has not been studied for medical students.

A review by Cornette et al. [6] examined each dimension of the IPTS in terms of its applicability to suicidal ideation or behavior among physicians and medical students and concluded that the literature provided strong support for the application of the IPTS in this population.

Concerning perceived burden, several circumstances may lead to feelings of burden. Medical trainees, as well as physicians, may experience academic failure or burnout, severe financial debt, emotional distress, mental disorders including burnout, increased sense of responsibility for patient’s disease, and professional conflict. For example, several studies reported that medical students have a higher risk of depression and that feelings of emotional burdensome are often secondary to depressive symptoms (e.g., self-defeat) [6].

Concerning thwarted belongingness, it has been suggested that the training environment may not provide high levels of support and that competition during medical studies can lead to a feeling of isolation and does not promote group membership. Some studies reported that medical students seek out friends or family for support instead of the specific services offered by their universities. Using professional services was considered to be potentially stigmatizing and harmful to future career [6]. Family members or friends may be unable to give adequate help for mental health needs and medical students may be left feeling isolated. However, one study reported that medical students who experienced suicidal ideations or behaviors more often used either maladaptive strategies, leading to mental and behavioral disengagement or were more likely to be indifferent towards their own or other’s suicidal behavior [7].

Concerning acquired ability, medical training, by definition, allows students to become less sensitive to pain and to become less emotionally reactive to injury and death. Moreover, medical trainees acquire knowledge about potentially lethal medications and several studies [6] reported that physicians were more likely to complete suicide by self-poisoning than individuals in the general population or in other specific groups.

Since this review, four studies [8,9,10,11] have explored the IPTS in physicians; however, it has not been studied in medical students. However, several authors have suggested that the IPTS could be also useful to explain the high rate of suicide in veterinarians [12]. Moreover, one study in a sample of 130 veterinary students reported that they became relatively fearless about death owing to their repeated exposure to euthanasia [13].

The present study aimed to explore the three constructs of the IPTS in medical students. Investigating medical students could make it possible to better understand the risk of suicide among physicians and enable the development of prevention strategies.

The present research has two specific aims. In line with the IPTS predictions, we tested the hypothesis that medical students with suicidality (ideators or planners) would have higher thwarted belongingness and perceived burdensomeness than medical students without suicidality. Further, we further whether acquired capability differentiated planners from ideators. In this study, depression, a potentially important confounding factor, was controlled.

## 2. Materials and Methods

### 2.1. Participants

The study participants were 178 undergraduate medical students (53 men, 125 women; 63 in the 2nd year of a Bachelor’s degree and 115 in the 3rd year of a Master’s degree) recruited from the Université Libre de Bruxelles (ULB). The study was approved by the Ethics Committee of the Hôpital Erasme. All students in the 2nd year of a Bachelor’s degree (N = 367) or the 3rd year of a Master’s degree (N = 244) were informed of the study via their email address. The participation was voluntary and the act of logging onto the online data collection system was used as informed consent. To participate in the study, the participants had to be French-speaking students aged 18 years and older. There were no exclusion criteria. The questionnaires were administered via an online data collection system.

### 2.2. Measures

The subjects filled out an ad hoc questionnaire and several rating scales.

The ad hoc questionnaire took 15–30 min to complete and contained 58 questions measuring work-related factors, health-related behavior, psychological distress, and demographic factors. Among the 59 questions, three evaluated the risk of suicide: “Have you ever had suicidal ideas?”, “Do you have a scenario, a specific suicidal plan?”, and “Have you ever attempted suicide?”.

Five versions of the Interpersonal Needs Questionnaire (INS) have been used in research, beginning with the original 25-item version [14]. Each of the proposed shorter versions (10, 12, 15, 18 items) draws a subset of items from the original 25-item version. One study has evaluated factor structure, internal consistency, and concurrent predictive validity of these five versions in three samples [15].

All versions had acceptable internal consistency and the 10- and 15-item versions had the best, most consistent model fit in the confirmatory factorial analyses. The authors recommended the use of the 15-item or 10-item versions. For the present study, we used the 15-item version. The INS [14] has 15 items (6 and 9 items measure thwarted belongingness (TB) and perceived burdensomeness (PB), respectively). Each item was scored from 1 (not at all true for me) to 7 (very true for me).

The Acquired Capability for Suicide Scale (ACSS, [16]) has 20 items measuring notable fearlessness of death or pain tolerance (7 items) and painful and provocative events (13 items). Each item was scored from 0 (not at all true for me) to 4 (very true for me). According to the authors [16], the 13 items measuring painful and provocative events are causal and not defining aspects of the acquired capability. In the present study, analyses were performed using either the 7 items measuring fearlessness of death or pain tolerance or the full 20-item version.

Depression was rated using the revised version of the Beck Depression Inventory (BDI, [17]). The BDI uses statements that best describe how the individual has felt during the previous two weeks. The French version of the BDI-II has satisfactory psychometric properties [18]. The total score ranges from 0 to 63 and a higher total score indicates more severe depressive symptoms.

For the present study, cognitive–affective symptoms of depression (CA-BDI) were assessed using the items relating to past failure, guilty feelings, punishment feelings, self-dislike, self-criticalness, and worthlessness. This subscale of the BDI has been used previously by Winer et al. [19].

In the present sample, the values of the Cronbach alpha coefficients of the French versions of the different rating scales were satisfactory, with values higher than 0.7 (PB: 0.85, TB: 0.84, ACSS: 0.77, BDI-II: 0.91).

Recent suicidal ideation (SID) was rated using the “Suicidal thoughts and wishes” item of the BDI-II in which 0 is “I don’t have any thoughts of killing myself” and 3 is “I would like to kill myself if I had the chance”. The response 0 was rated “No recent suicidal ideas” and the other responses were rated “Recent suicidal ideas”.

### 2.3. Statistical Analyses

Based on their answers to the ad hoc questionnaire and to the suicidal thoughts and wishes item of the BDI-II, the medical students were classified as having no suicidality, lifetime ideators, recent ideators, or planners. The independent variable was the group (no suicidality, lifetime ideators, recent ideators, and planners). The covariates were age, gender, and the cognitive subscore of the BDI-II (CA-BDI). The dependent variables were the rating scales scores: TB, PB, and ACSS. We reported the results concerning the full version of the ACSS (20 items), mentioning the results of the short version (7 items measuring fearlessness of death or pain tolerance) only if they were different from the full version.

First, bi-variate analyses using Chi-Square analysis and one-way analyses of variance (ANOVA) were performed using group as the inter-subject variable and each dependent variable or covariate.

Second, multivariate analyses using analyses of covariance (ANCOVA) were performed using group as the inter-subject variable, each significant covariate found in ANOVA, and each significant dependent variable found in ANOVA.

## 3. Results

There were 95 subjects without suicidality, 24 subjects with lifetime suicidal ideation and without recent suicidal ideation, 28 subjects with recent suicidal ideation (all had lifetime suicidal ideation), and 26 planners (with or without lifetime or recent suicidal ideation). Five subjects with previous suicide attempts were excluded (Table 1). The percentages in the different groups did not differ significantly between the medical students in the 2nd year of a Bachelor’s degree or in the 3rd year of a Master’s degree (χ^2^ = 4.26, df = 3, *p* = 0.235).

### 3.1. Bivariate Analyses

There were significant differences between the groups for the CA-BDI, but the four groups did not differ by gender and age. For the dependent variables, the four groups had significant differences for the TB and PB scores but not for the ACSS score (Table 1).

### 3.2. Multivariate Analyses

Analyses of covariance (ANCOVA) was performed using the groups as independent variable, CA-BDI as the covariate, and PB or TB as dependent variables.

For TB, ANCOVA was significant (F (4, 168) = 11.13, *p* < 0.0001) and post hoc tests (Fisher LSD) were performed by comparing each group pairwise. There were no significant differences in the following comparisons: 1, between subjects without suicidality and subjects with lifetime suicidal ideation (*p* = 0.62); 2, between subjects with recent suicidal ideation and planners (*p* = 0.07); and 3, between subjects with recent suicidal ideation and subjects with lifetime suicidal ideation (*p* = 0.09). Subjects with recent suicidal ideation had significantly higher scores of TB than subjects without suicidality (*p* = 0.007). Planners had significantly higher scores of TB than subjects without suicidality (*p* = 0.0001) or subjects with lifetime suicidal ideation (*p* = 0.0001). Thus, high levels of TB were characteristic of subjects with recent suicidal ideation and planners compared with subjects without suicidality.

For PB, ANCOVA was significant (F (4, 168) = 9.78, *p* < 0.0001) and post hoc tests (Fisher LSD) were performed by comparing two by two each group. There were no significant differences between subjects without suicidality and subjects with lifetime suicidal ideation (*p* = 0.66) or subjects with recent suicidal ideation (*p* = 0.52). There was no significant difference between subjects with lifetime suicidal ideation and subjects with recent suicidal ideation (*p* = 0.89). Planners had significantly higher scores of PB than subjects without suicidality (*p* = 0.0001), subjects with lifetime suicidal ideation (*p* = 0.001), or subjects with recent suicidal ideation (*p* = 0.001). Thus, high levels of PB were characteristic of planners compared with the three other groups.

## 4. Discussion

To the best of our knowledge, no study has explored the IPTS in medical students. The results of the present study can be compared either with the results of studies exploring the three constructs of the IPTS in physicians, in other health professionals (e.g., veterinarians), or in university health students. No studies in medical students, in physicians, in other health professionals, or in university health students have compared PB, TB, or ACSS between subjects without suicidality and subjects with suicidal ideations or suicide plans. Moreover, our results can be compared with the results of studies comparing TB, PB, or ACSS in various samples of subjects without suicidal ideation, with suicidal ideation, or with suicide plans.

Concerning the first hypothesis that ideators or planners would have higher perceived burdensomeness and thwarted belongingness than medical students without suicidality.

In 2017, Forrest and Smith [20] compared the scores of 106 subjects recruited on an online survey platform: 25 subjects did not have lifetime suicidal ideation, plans, or attempts, or current suicidal ideation; 81 had lifetime suicidality (36 with lifetime or current suicidal ideations, 15 with plans, and 30 with at least one lifetime suicide attempt). The four groups were compared for TB and PB. Significant differences among the groups were found for TB and PB. Then, the authors compared the subjects without suicidality (*n* = 25) with the subjects with suicidality (*n* = 81) and significantly higher scores were found on the TB and PB subscales in the subjects with suicidality. Moreover, neither ideators nor planners had significant differences in TB and PB and each group had significantly higher scores for the two rating scales compared with subjects without suicidality. Unfortunately, the authors did not distinguish between lifetime ideators and recent ideators and the potential effect of depression was not controlled.

For TB, the results of Forrest and Smith [20] partially confirmed those of the present study, reporting no significant differences between planners and recent ideators. For PB, contrary to the results of the present study, Forrest and Smith [20] reported that ideators and planners had no significant differences. The explanation of the difference between the two studies could be related to the potential effect of depression, as depressive level was not controlled in the Forrest and Smith study, and the absence of distinction between lifetime and recent ideators.

There have been two systematic reviews of the predictions of the IPTS. For the relationship between TB and suicidal ideation, the first review [21] examined 55 studies; in 22 (40%), the relationship was significant, and in 33 (60%), it was not. The second review [22] found that TB was significantly and moderately related to suicidal ideation (r = 0.37, *p* < 0.001; k (number of samples) = 84; N = 37,952). Concerning the relationship between PB and suicidal ideation, the first review [21] examined 69 studies; in 57 (82.6%), the relationship was significant, and in 12 (17.3%), it was not significant. The second review [22] found that PB was significantly and moderately related to suicidal ideation (r = 0.48, *p* < 0.001; k = 84; N = 37,894).

Concerning the second hypothesis that planners would have higher acquired capability than ideators.

The present study did not find significant difference on the ACSS between the four groups of subjects. In the study of Forrest and Smith [22], there was no significant difference on the ACSS between their four groups.

In the first systematic review [21], there were 21 studies of the relationship between ACSS and suicidal ideation; in 12 studies, the relationship was significant (57.1%) and in 9 studies (42.8%), it was not. In the second review [22], ACSS was significantly and weakly related to suicidal ideation (r = 0.10, *p* < 0.02; k = 29; N = 9782).

## 5. Conclusions

The present study had several limitations. First, taking into account the low number of subjects in several groups, negative results could be explained by low statistical power of the statistical tests. Second, the results must be confirmed in larger groups of medical students from different universities. Third, it could be interesting to test the three characteristics of the IPTS in a sample of medical students with lifetime and/or recent suicide attempts. Fourth, the ad hoc questionnaire does not make it possible to specify the previous suicide attempts, particularly if they occurred during medical studies. Fifth, gender and education levels are possible biases.

The results of the present study support the notion that, among the three characteristics of the IPTS, perceived burdensomeness could be a predictor of high risk of suicide in medical students, even when depression was controlled. This result must be confirmed as it may allow the detection of high-risk subjects. The current study strongly suggests the value of filling a systematic questionnaire (INS and BDI-II) for each year of medical study. Future research could explore the relationship between perceived burdensomeness and burnout or anhedonia.

## Figures and Tables

**Table 1 ijerph-18-11526-t001:** Demographic and psychometrical variables in the different groups.

	No Suicidality (*n* = 95)	Life-Time Ideators (*n* = 24)	Recent Ideators (*n* = 28)	Planners (*n* = 26)	Statistical Tests (Chi-Square Test (χ^2^) or Analysis of Variance)
Gender (Women %)	64 (67.4%)	19 (79.2%)	22 (78.6%)	17 (65.4%)	χ^2^ = 2.52, df = 3, *p* = 0.472
Age < 25 years (%)	77 (81%)	16 (66.7%)	19 (73.1%)	22 (78.6%)	χ^2^ = 2.61, df = 3, *p* = 0.46
Perceived burdensomeness (PB)	7.6 (4.64)	8.04 (3.1)	8.21 (3.52)	12.19 (6.6)	F (3, 169) = 6.73, *p* < 0.001
Thwarted Belongingness (TB)	28.77 (11.3)	29.96 (9.4)	34.89 (12.07)	40.08 (10.16)	F (3, 169) = 8.22, *p* < 0.001
Acquired Capacity for Suicide (ACSS)	38.93 (12)	37.92 (9.22)	36.71 (8.57)	41.08 (12)	F (3, 169) = 0.74, *p* = 0.53
CA-BDI	2.98 (2.77)	3.71 (2.84)	5.61 (3.67)	7.42 (4.03)	F (3, 169) = 15.88, *p* < 0.001

(CA-BDI: cognitive–affective components of the Beck Depression Inventory).

## Data Availability

Requests for access will be reviewed by the corresponding author.

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
