# Peer review of "The Interpersonal–Psychological Theory of Suicide in Medical Students: Comparisons of Individuals without Suicidality, Ideators, and Planners"

_ijerph, 2021, doi:10.3390/ijerph182111526_

Round 1

Reviewer 1 Report

Abstract - Please include the abbreviation after the first mention of the theory.

The authors have used IPT in the abstract, but IPTS in the main text - please change this to be consistent. 

Also, it would be good if more of a description of the sample could be included in the abstract. 

Introduction -  The frequency of depressive symptoms and suicidal ideation is given for medical students, but it would also be helpful to include something about suicide rates in this population. 

The rationale for why this is an important group to investigate could be more explicit. 

The paragraph on perceived burden is more about stress on the students rather than feeling like they are a burden on others, and I feel this needs further work to link it more explicitly to the theory.  

Discussion - line 210-212 - the study in Mexico does not seem relevant to the discussion, I'd suggest removing it. 

Some of the information on previous studies seems like it would be better placed in the introduction, as it reads as quite descriptive in places for a discussion. 

The authors could consider making suggestions for future research and also consider the implications of the current study more. 

Author Response

Responses to the reviewer 1

  • In the abstract the abbreviation (IPTS) is included after the first mention of the theory.
  • IPTS is used in the abstract as well as in the main text.
  • A description of the sample is included in the abstract (see lines 15-16).
  • Suicide rates in medical students are mentioned (see lines 36-37) in the Introduction with a new reference (see reference 2).
  • The rationale of the study is more precised (see lines 91-92).
  • I agree with the remark of the reviewer. Perceived burden could be secondary to depression and notably an expression of cognitive symptoms (self-depreciation…) (see lines 67-68).
  • The study in Mexico has been deleted.
  • I have kept the previous studies in the discussion section. To the best of our knowledge there is no another studies on IPTS in medical students and thus to discuss our results we have mentioned studies including students of health universities (veterinarians).
  • Implications of the current study and suggestions for future research are presented (see lines 287-290).

Reviewer 2 Report

The manuscript addresses a relevant issue in relation to suicidal behaviors in medical students.

 the introduction made is appropriate and relevant to the topic. This allows us to recognize the relevance of studying suicide risk in medical students from IPTS.

The aim is too general, since it does not include the comparison between groups with different suicidal behaviors.

In the methodological part, there are a series of limitations.

The percentage of women in the sample far exceeds the percentage of men. The study incorporates undergraduate medical students and master’s degree students, the last one is being most of the study participants. The evaluation of suicidal behaviors by means of ad hoc questions is not precise since it does not allow to clearly establish whether planning suicide attempt has occurred throughout life, during the medicine’s studies, the last 12 months or in the actual moment.  

The procedure for inviting and selecting participants is unclear. It is necessary to report the total number of medical students in the ULB.

Cronbach's alpha was not assessed in the ad hoc questionnaire.

When making multiple comparisons, it is advisable to use some correction procedure.

In table 1, integrate the statistical data into the table as one more column.

The discussion is not clear as to the aim of the study as it does not make a series of inferences regarding the presence of TB and PB in the groups that present recent suicidal ideation in contrast to suicidal ideation at some time in life without suicidal ideation in students of Medicine.

The possible biases due to the characteristics of the sample such as gender and educational level are not pointed out in the limitations.

Author Response

Responses to the reviewer 2

  • The covariable gender was controled (see line 183). Educational level was also controled (see lines 171-174) and a new CHI square test was done. The use of the ad hoc questionnaire is mentioned as a limitation (see lines 281-282).
  • The procedure for inviting and selecting participants is more precised mentioning the total number of students in 2nd year Bachelor’s degree (N = 367) or in 3rd year Master’s degree (N = 244) (see lines 103-105).
  • Cronbach alpha was not assessed as the variables are nominal and not continuous.
  • Post-hoc tests following ANOVA or ANCOVA were Fisher LSD that included some correction procedure (Alpha type 1 risk).
  • In Table 1, statistical data are included .
  • Gender and educative levels are controled covariables and could be possible biases (see line 283).

Round 2

Reviewer 2 Report

 I appreciate the improvements in the manuscript. Nevertheless, The discussion is unchanged. I consider that the discussion is not clear in relation to the aims and the main findings.

“We tested the hypothesis that medical students with suicidality (ideators or planners) would have higher thwarted belongingness and perceived burdensomeness than medical students without suicidality. We further tested whether acquired capability differentiated planners from ideators. "

It is not discussed why the findings of the highest levels in TB is an indicator associated with recent suicidal ideation and the current suicide plan and not with life-time suicidal ideation. It is not discussed why the highest levels in PB were only in planners. The lack of differentiation based on ACSS between participants is not discussed. I insist that the discussion should not be organized by each referred study, instead of based on the objectives and findings.

It is necessary to adjust the format of the table according to the first version (it looked better).

Author Response

Responses to the reviewer 2

The discussion has been reorganized as requested by the reviewer. The two hypotheses are discussed and it is noted that only one study is available for discussion (Forrest & Smith, 2017). In addition, two meta-analyzes are cited (Chu et al, 2017 ; Ma et al, 2016).

Round 3

Reviewer 2 Report

I appreciate the improvements in the manuscript.

Best regards

This manuscript is a resubmission of an earlier submission. The following is a list of the peer review reports and author responses from that submission.